# Pharmacokinetics of Moxidectin combined with Albendazole or Albendazole plus Diethylcarbamazine for Bancroftian Filariasis

Yashpal S. Chhonker[1], Catherine Bjerum[2], Veenu Bala[1], Allassane F. Ouattara[3], Benjamin G. Koudou[3], Toki P. Gabo[4], Abdullah Alshehri[1], Abdoulaye Meïté[5], Peter U. Fischer[6], Gary J. Weil[6], Christopher L. King[2,7], Philip J. Budge[6], Daryl J. Murry[1,8]*

1 Clinical Pharmacology Laboratory, Dept of Pharmacy Practice and Science, University of Nebraska Medical Center, Omaha, Nebraska, United States of America, 2 Center for Global Health and Diseases, Case Western Reserve University School of Medicine, Cleveland, Ohio, United States of America, 3 Centre Suisse de Recherche Scientifique en Côte d'Ivoire (CSRS), Abidjan, Ivory Coast, 4 Centre Hospitalier Regional d'Agboville, Côte d'Ivoire, 5 Programme National de la Lutte Contre la Schistosomiase, les Geohelminthiases et la Filariose Lymphatique, Ivory Coast, 6 Infectious Diseases Division, Department of Medicine, Washington University School of Medicine, St. Louis, Missouri, United States of America, 7 Veterans Affairs Research Service, Cleveland Veterans Affairs Medical Center, Cleveland, Ohio, United States of America, 8 Fred and Pamela Buffett Cancer Center, University of Nebraska Medical Center, Omaha, Nebraska, United States of America

* dj.murry@unmc.edu

**Data Availability Statement:** Data contained within the article and any supplementary data file.

## Abstract

Moxidectin (MOX) is a milbemycin endectocide recently approved by the U.S. FDA for the treatment of onchocerciasis in persons at least 12 years of age. MOX has been shown to have a good safety profile in recent clinical trials. The efficacy of MOX for the treatment of lymphatic filariasis (LF) and its potential use in mass drug administration protocols for the elimination of LF is currently under evaluation. In the context of a clinical trial, we investigated the pharmacokinetics and drug interactions of a combination of MOX plus albendazole (ALB) with or without diethylcarbamazine (DEC) compared to ivermectin (IVM) plus ALB with or without DEC in the following four different treatment arms: (I) IVM (0.2mg/kg) plus DEC (6 mg/kg) and ALB (400mg); (II) IVM plus ALB; (III) MOX (8 mg) plus DEC and ALB; and (IV) MOX plus ALB. Drug concentrations were determined using validated liquid chromatography-mass spectrometric methods. Pharmacokinetic parameters were determined using standard non-compartmental analysis methods. Statistical analysis was performed using JMP software. Fifty-eight of 164 study participants (53 men and five women) were included with ages ranging from 18 to 63 yrs (mean = 37). MOX apparent oral clearance (Cl/F) ranged from 0.7 to 10.8 L/hr with $C_{max}$ values ranging from 20.8 to 314.5 ng/mL. The mean (range) area under the curve $(AUC)_{0-\infty}$ for MOX, 3405 ng*hr/mL (742–11376), and IVM 1906 ng*hr/mL (692–5900), varied over a ~15.3 and ~8.5-fold range, respectively. The geometric mean ratio for $C_{max}$, $AUC_{0-t}$, and $AUC_{0-\infty}$ were within the no-drug interaction range of 80–125% for all drugs. This indicates that the addition of MOX to ALB alone or ALB plus DEC for LF therapy did not alter the drug exposure of co-administered drugs compared to IVM combinations.

**Clinical Trial Registration**: NCT04410406, https://clinicaltrials.gov/.

**Funding:** This work was financially supported by the Bill and Melinda Gates Foundation (https://www.gatesfoundation.org/ [nam10.safelinks.protection.outlook.com]) with grant OPP1201530 (GJW). PJB was supported in part by a NIH/NIAID career development grant K08AI121422.The funders had no role in study design, data collection and analysis, decision to publish, or preparation of the manuscript.

**Competing interests:** The authors have declared that no competing interests exist

## Author summary

Mass Drug administration (MDA) with multiple drugs is common for the treatment and control of lymphatic filariasis (LF). This study aimed to determine whether the incorporation of moxidectin (MOX) into MDA regimens for LF will alter the pharmacokinetics (PK) of co-administered drugs. In the context of a clinical trial, we investigated the PK and drug interactions of a combination of moxidectin (MOX) plus albendazole (ALB) with or without diethylcarbamazine (DEC) compared to ivermectin (IVM) plus ALB with or without DEC. The study was conducted in adult patients located in Côte d'Ivoire. The addition of MOX did not alter the exposure (AUC0-t) or maximum concentration (Cmax) of co-administered drugs. These results suggest the incorporation of MOX into current MDA LF treatment programs will not result in clinically significant alterations in drug exposure.

## Background

Lymphatic filariasis (LF) is a nematode infection, caused by *Wuchereria bancrofti*, *Brugia malayi or B. timori*, and 51.4 million people are worldwide infected. LF infection leads to lymphatic system dysfunction, which causes clinical manifestations of the disease, such as lymphedema that can progress to elephantiasis [1]. The disease is associated with disability, and social stigma [2]. Global efforts to eliminate LF transmission have reduced the population at risk for the disease, but LF transmission continues in at least 53 countries with 886 million still at risk [3–5]. In 2000, the WHO established the Global Program to Eliminate Lymphatic Filariasis (GPELF) to stop the transmission of infection by mass drug administration (MDA) of anthelmintics. [3]. Since the start of GPELF, the number of infections has been reduced by 74% globally. Based on the current progress in disease elimination efforts, WHO has proposed a target to reduce infections by 90% from all remaining endemic areas by 2030[6]

The mass drug administration (MDA) program is based on once or twice a year therapy for all eligible persons in endemic areas to decrease the prevalence of infection in the community to the level that cannot support the transmission of the new infection [4,7]. The most recent summary of MDA results from WHO reported that since 2000, more than 9 billion cumulative treatment regimens have been delivered by MDA to more than 935 million people, [7], making GPELF the largest MDA-based infectious disease intervention program attempted to date. The MDA treatment medications utilized are diethylcarbamazine plus albendazole and ivermectin known as IDA, and albendazole in combination with diethylcarbamazine (2-Drug therapy; DA). The IDA treatment protocol has shown efficacy against LF infection by the sustained suppression of microfilaremia in the residents of West Africa and the South Pacific [8–11].

The triple drug combination therapy with all three medications (IDA) was recently found to be superior to the 2-drug combinations in both Papua New Guinea (DA) and Côte d'Ivoire (IA) [8,11,12]. IDA triple therapy produces prolonged Mf clearance after only one dose with results superior to those obtained after either DA or IA suggesting that MDA with IDA might achieve interruption of transmission with 3 or fewer rounds of MDA [13]. Large-scale international community safety trials in over 20,000 participants have shown no increase in adverse events (AEs) with IDA compared to DA, and thus IDA has been recommended by WHO for MDA in areas of the world without onchocerciasis or loiasis [14].

DEC is not used for MDA in LF/onchocerciasis co-endemic areas, because it can cause serious ocular AEs and life threatening systemic reactions in persons with intraocular *Onchocerca*

*volvulus* Mf. Neither DEC nor IVM are safe for routine MDA in areas with loiasis ("African eye worm") caused by *Loa loa*, since severe AEs (SAEs) including encephalopathy and death have occurred in DEC/IVM treated persons with very heavy loiasis (>20,000 *L. loa* Mf/mL in peripheral blood) infections. This risk of SAEs in loiasis-endemic African countries has delayed effective MDA in these areas and remains a major challenge to LF and onchocerciasis elimination in Africa. Other challenges facing the GPELF include the limited macrofilaricidal activity of current MDA regimens, which necessitates repeated annual rounds of MDA, and the difficulty of achieving high compliance rates for MDA over a period of years [15,16].

The purpose of the current study is to determine whether the incorporation of moxidectin (MOX) into MDA regimens for LF will alter the pharmacokinetics of co-administered drugs. MOX was approved in June 2018 by the U.S. FDA for the treatment of onchocerciasis [17]. Its mode of action is similar to that of IVM, and a single treatment dose does not kill adult *O. volvulus* worms but is superior to IVM for achieving prolonged clearance of microfilariae (Mf) from the skin [18]. For moxidectin, PK properties have been assessed only in a limited number of studies [19,20]. Recent clinical trials have shown that the combination of moxidectin/albendazole might be more active than MOX alone in soil borne helminth infections [21,22]. Another trial has been conducted to explore efficacy and safety of moxidectin and albendazole compared with ivermectin and albendazole coadministration in an adolescent population infected with *Trichuris trichiura* in Tanzania [23]. No clinical trials have been reported to date to assess the pharmacokinetics (PK) of MOX (alone or in combination with DEC/ALB) in adult populations for the treatment of LF.

Pharmacokinetic (PK) data are needed to ensure that there are no significant drug-drug interactions that might impact either the safety or efficacy of co-administration of the combination of MOX plus ALB with or without DEC compared to IVM plus ALB with or without DEC. There is no PK data on the co-administration of MOX with or without the addition of DEC or ALB. We, therefore, conducted a study to provide the first data on the PK of MOXDA and MOXA vs. IDA or IA for treating bancroftian filariasis.

## Methods

### Study setting and ethical statement

This trial was conducted at the Centre de Recherche de Filariose Lymphatique d'Agboville, located at the Centre Hospitalier Regional (CHR) d'Agboville, Côte d'Ivoire, and in the surrounding communities. The study was performed in coordination with the national LF elimination program's regional assessment of LF transmission. All individuals provided written informed consent to participate in the study. The study protocol and related documents were approved by the institutional review boards in St. Louis (Washington University, IRB#202005076), Cleveland, USA (Case Western Reserve University, IRB#STUDY20200714) and in Cote d'Ivoire (CNESVS #011-20/MSHP/CNESVS-km). This trial is registered at https://clinicaltrials.gov/ (NCT04410406).

Eligible participants were adults (men and women aged 18–70 years) with *W. bancrofti* infection and ≥ 40 Mf/mL venous night blood, residing in or near Agboville district, Côte d'Ivoire. The study was amended to include participants with *W. bancrofti* infection counts ≤ 40 Mf/mL venous blood. At enrollment, participants underwent a standardized medical examination and blood and urine tests. Exclusion criteria included alanine transaminase (ALT), aspartate transaminase (AST), or creatinine values greater than >2 times the upper limit of normal; hemoglobin levels less than 7 gm/dL; greater than 2+ proteinuria or hematuria by urine dipstick, pregnancy, or a positive skin snip result for onchocerciasis.

## Source of medications and treatment

MOX was generously provided by Medicines Development for Global Health (MDGH). Diethylcarbamazine (Banocide GlaxoSmithKline), Albendazole (Pfizer, India), and Ivermectin (Merck & Co. Inc.) were purchased for the study. A fixed dose of 400 mg for albendazole and 8 mg for MOX was used for all participants. Ivermectin and DEC doses were 200 μg/kg and 6 mg/kg, respectively.

## Randomization and masking

Eligible participants were randomly assigned by use of a computer generated randomization sequence stratified by sex to receive one of the treatment regimens: (ARM-I, IDA) IVM (0.2mg/kg) plus DEC- citrate (6 mg/kg) and ALB (400mg); (ARM-II, IA) IVM plus ALB; (ARM-III, MOXDA) MOX (8 mg) plus DEC and ALB; and (ARM-IV, MOXA) MOX plus ALB.

## Blood collection and Bioanalysis

All participants received a typical Ivorian breakfast of wheat bread and eggs. Participants were treated approximately 30 minutes after the meal with co-administered doses as described above. Venous blood samples were collected at 0, 2, 3, 4, 6, 8, 12, 24, 36, 48, and 72 hours, and at 7 days after treatment, with aliquots of plasma stored at -20˚C on site for up to 4 weeks, then at -80˚C until analysis.

Plasma concentrations of DEC, ALB, albendazole sulfoxide (ALB-OX), and albendazole sulfone (ALB-ON) were determined using a validated liquid chromatography-mass spectros-copy (LC-MS/MS) method [24]. For IVM, plasma concentrations were determined using an LC-MS/MS assay as previously described [25]. MOX plasma concentrations were determined using a sensitive LC-MS/MS assay as previously described [26].

## Pharmacokinetic methods

Chemical and materials. Pharmaceutical grade DEC, IVM, ALB, ALB-OX, ALB-ON, oxiben-dazole (OBZ), MOX, and deuterated diethylcarbamazine (D3-DEC) were obtained from Sigma-Aldrich, St Louis, MO, USA. Ultrapure water was obtained from a Barnstead Ultrapure Thermo-Scientific water purification system. Methanol, tetrahydrofuran, acetic acid, formic acid, and acetonitrile were purchased from Thermo Fisher Scientific (Fair Lawn, NJ, USA). Centrifuge tube filters were obtained from Corning Co. (Corning, NY, USA). Agilent (Santa Clara, CA, USA) bond Elute C18 solid phase extraction cartridges (50mg/mL) were used to extract drugs from plasma. All other chemical reagents were purchased from Thermo Fisher Scientific and were of analytical grade.

## Liquid chromatographic and mass spectrometric conditions

Plasma concentrations of DEC, ALB, ALB-OX, ALB-ON, IVM, and MOX were determined using our previously reported validated bioanalytical LC-MS/MS methods (18–20). These methods were validated according to the guidelines of US Food and Drug Administration (FDA-2018) for industry bioanalytical method validation. The both assays were linear concentration range of 0.1 to 1000ng/mL for MOX and IVM, 1–2000 ng/mL for DEC, 0.5–1000 ng/mL for ALB-OX, 0.1–200 ng/mL for ALB and ALB-ON. At least 75% of the standards were within the acceptable limits within ±15% of the nominal concentration except at LLOQ where should not deviate by more than 20% for acceptance of the generated calibration curve.

Dilution integrity was validated for each analyte. These data confirm that the method described has a satisfactory accuracy and precision for the quantitation of all the analytes.

Blood samples for PK analysis were shipped on dry ice to the University of Nebraska Medical Center. The PK parameters of DEC, ALB, ALB-SOX, ALB-SON, IVM and MOX were calculated by non-compartmental analysis (NCA) using Phoenix WinNonlin-8.3 (Certara, Princeton, NJ, USA). The maximum concentration ($C_{max}$), and time to reach $C_{max}$ ($T_{max}$) were determined directly from the plasma concentration-time data. The area under the curve ($AUC_{0-inf}$), was estimated using the trapezoidal method from 0 to $t_{last}$ and extrapolation from $t_{last}$ to infinity ($AUC_{0-\infty}$) based on the observed concentration at the last time point divided by the terminal elimination rate constant ($\lambda_z$). The half-life (t ½) was calculated using the formula of $0.693/ \lambda_z$. The apparent volume of distribution (Vz/F) and clearance (CL/F) for each drug was calculated using standard equations. Values of $C_{max}$, $AUC_{0-t}$, and $AUC_{0-\infty}$ were dose normalized to reduce PK parameters variability resulting from per kg doses administered to each subject.

The primary outcome was lack of clinically relevant pharmacokinetic drug interactions, defined as geometric mean ratios (GMRs) within the conventional acceptance range of 80–125% for the $C_{max}$, $AUC_{0-t}$, and $AUC0_{-\infty}$ between treatment arms. GMR was used as previous studies have shown the pharmacokinetics of IVM, DEC, and ALB to be highly variable (CV% greater than 30%) [11,27].

## Statistical analysis

The statistical analysis was performed on dose-normalized and natural log-transformed MOX exposure parameters ($AUC_{0-t}$ and $C_{max}$), in each study arm. P-values of $<0.05$ were deemed to indicate statistical significance. Summary statistics of the pharmacokinetic parameters were conducted using JMP software (SAS Institute, Cary NC; version 14.3.0). Power calculations indicated that 58 participants (subdivided in 14 or 15 subjects per arm) would give a power of 80% to test the hypothesis that the primary outcome of bioequivalence between test groups between 80–125% of geometric mean ratio (see below) based on previous PK modeling studies [12] and European Medicines Agency guidelines [28] with the assumption that 10% of participants would be lost to follow-up. For analysis of the primary outcome (lack of clinically relevant pharmacokinetic interactions), we estimated one-sided 90% CI for the geometric mean ratios (GMRs) of the experimental regimen and the reference regimens. Descriptive comparisons of PK parameters between arms were performed using the Kruskal-Wallis test using the JMP software. 90% CI was estimated for each pharmacokinetic parameter, after log transformation using Phoenix WinNonlin-8.3 (Certara, Princeton, NJ, USA). The data obtained in this study were evaluated according to Food and Drug Administration and European Medicines Agency guidelines (EMA) for assessment of a significant drug interaction [28,29]. According to the EMA guideline, the wider equivalence range could be considered for highly variable drugs (intra-subject coefficient of variation $> 30\%$). Previous studies have shown the substantial PK variability with a coefficient of variations for $AUC_{0-\infty}$, $AUC_{0-t}$, and $C_{max}$ greater than 30% for DEC, IVM, and $C_{max}$ greater than 50% for ALB and its active metabolite [28,30].

## Results

### Study population

Fifty-eight of 164 study participants (53 men and five women) were participated in the pharmacokinetic sub-study. Baseline demographics were similar across all treatment groups and representative of the study population. Patient demographics for each study arm are presented in Table 1. Mean (SD) age was 38.7 (13.6), 34.3 (9.8), 39.9 (10.4) and 35.1 (11.3) years for the

**Table 1. Baseline demographic and clinical characteristics, Mean (±SD).**

| Parameters | Arm 1: IDA | Arm 2: IA | Arm 3: MOXDA | Arm 4: MOXA |
|---|---|---|---|---|
| | Mean (SD), n = 15 | Mean (SD), n = 15 | Mean (SD), n = 14 | Mean (SD), n = 14 |
| Age (years) | 38.7 (13.6) | 34.3 (9.8) | 39.9 (10.4) | 35.1 (11.3) |
| Sex (M: F) | 14:1 | 13:2 | 13:1 | 13:1 |
| Weight (Kg) | 63.4 (10.5) | 65.1 (7.7) | 59.4 (7.3) | 61.5 (7.9) |
| Height (cm) | 169.4 (8.3) | 166.4 (10.8) | 161.2 (25.7) | 169.4 (6.3) |
| Height (inch) | 66.7 (3.3) | 65.5 (4.3) | 63.5 (10.1) | 66.7 (2.5) |
| Dose DEC-Citrate given (mg) | 360 (73.7) | NA | 350 (51.9) | NA |
| Dose ALB given (mg) | 400 | 400 | 400 | 400 |
| Dose IVM given (mcg/kg) | 200 | 200 | NA | NA |
| Dose MOX given (mg) | NA | NA | 8 | 8 |
| BSA (M2) | 1.7 (0.2) | 1.7 (0.2) | 1.6 (0.2) | 1.7 (0.1) |
| BMI | 21.9 (2.1) | 23.6 (2.5) | 26.7 (21) | 21.3 (1.3) |
| IBW (KG) | 65.1 (8.1) | 62.6 (9.5) | 59.8 (16.1) | 65.1 (6) |
| ABW (KG) | 62.3 (9.4) | 62.4 (8.1) | 56.8 (10.2) | 61.3 (7.5) |
| LBW (KG) | 50.7 (7.6) | 50.2 (7.7) | 46.2 (9.8) | 49.8 (6.3) |
| CrCl (ml/min) | 77.5 (30.2) | 79.7 (29.4) | 77.9 (15.3) | 94.8 (37.6) |
| Cr (mg/dl) | 2 (3.6) | 1.9 (3.2) | 1 (0.2) | 1.9 (3.7) |
| ALT (u/L) | 24.9 (6.6) | 31 (10) | 27.4 (8.7) | 26.9 (8.7) |
| AST(u/L) | 35.8 (9.5) | 43.5 (15.4) | 40.5 (12.3) | 37.9 (8.9) |

IDA, IA, MOXDA, and MOXA Arms, respectively. Mean (SD) weight was 63.4 (10.5), 65.1 (7.7), 59.4 (7.3) and 61.5 (7.9) for the IDA, IA, MOXDA, and MOXA arms, respectively. The number of women was 1 of 15, 2 of 15 and 1 of 14 and 1 of 14 for the IDA, IA, MOXDA, and MOXA Arms, respectively.

## Pharmacokinetics drug-drug Interactions

The mean plasma concentration–time profiles of ALB-OX (the active metabolite of ALB) DEC, IVM and MOX are shown in **Fig 1**. PK parameters for IDA (Arm-I), IA (Arm-II), MOXDA (Arm-III) and MOXA (Arm-IV) are shown in **Table 2 and S1 Table**. The Mean $T_{1/2}$ and $T_{max}$ and pharmacokinetic exposure parameters ($AUC_{0-t}$ and $C_{max}$) for treatment drugs, were not different between study arms (Table 2). The mean (range) area under the curve ($AUC0-\infty$) for MOX, 3406 (742–11376), and IVM 1906 (692–5900), varied over a ~15.3 and ~8.5-fold range, respectively. MOX apparent oral clearance (CL/F) and volume of distribution (V/F); mean (range) for all subject, was 3.6 (0.7–10.8) L/h and 410.8 (78.6–1088.5) L, respectively. The mean (range) for all subject, MOX area under the curve $(AUC)_{0-\infty}$ was 3404.7 (742.2–11375.5), and IVM $AUC_{0-\infty}$ was 1905.9 (692.3–5900.1) and varied over a ~15.3 and ~8.5-fold range, respectively. Median values were not different between study arms (p>0.05) **S1 Table**.

Dose-normalized $C_{max}$ and $AUC_{0-t}$ are presented as box-whisker plots of ALB-OX (the active metabolite of ALB) DEC, IVM and MOX by study arms in **Fig 2**. Distribution of dose adjusted $C_{max}$ and $AUC_{0-t}$ of study drugs by study arm with individual data points were not significantly different (P>0.05) when comparing participants in different arms.

Geometric mean ratios (GMR) of parameters in the experimental arm (IDA or MOXDA) versus the reference arms (IA or MOXA) are presented with 90% confidence intervals (CIs) in **Fig 3**. $C_{max}$, $AUC_{0-t}$, and $AUC_{0-\infty}$ for each analyte were dose-normalized. The GMR of $C_{max}$, $AUC_{0-t}$, and $AUC_{0-\infty}$ for DEC, IVM, and MOX were within the range of 80–125%, and the

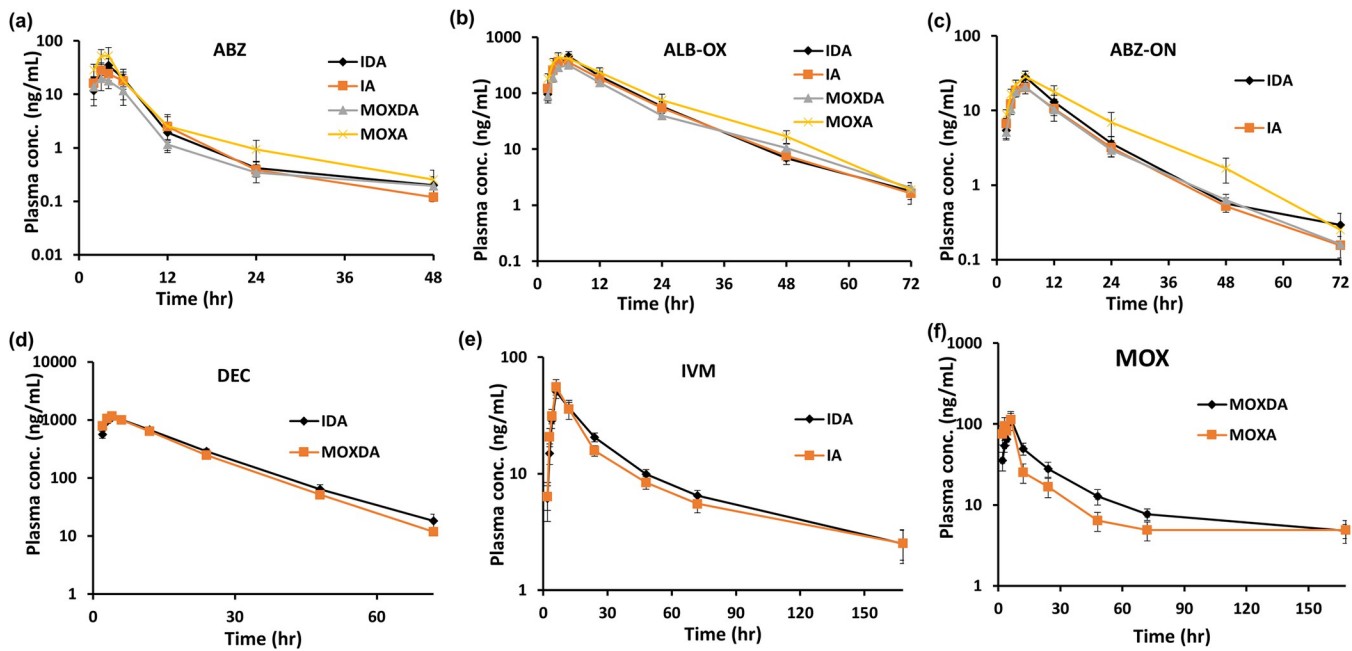

**Fig 1. Drug concentration vs time curve plots for subjects on the IDA, IA, MOXDA and MOXA study arms.** Overlay of mean (±SEM) plasma concentration-time profiles of (a) **ALB**, (b) **ALB -OX** (c) **ALB-ON**, (d) **DEC (e) IVM, and (f) MOX** after a single dose separated by study arm (IDA, n = 15, IA, n = 15, MOXDA, n = 14, MOXA, n = 14).

90% CIs partly overlap the range of 50–200% that reflects the inter subject variability. For ALB, which is rapidly metabolized to ALB-SOX, the GMR of $C_{max}$, $AUC_{0-t}$, and $AUC_{0-\infty}$, were within the range of 80–125%. (

## Discussion

The objective of this study as to characterize the PK and drug-drug interactions of MOX after a single oral dose (8 mg fixed in *W. bancrofti*-infected subjects) in combination with ALB with or without DEC as a part of a clinical trial for the treatment of LF. Following a single dose, MOX concentrations rapidly reached a peak concentration in 5 hours. Following the peak concentration there is a very pronounced distribution phase where plasma levels decline rapidly by approximately 24 hours, followed by a prolonged elimination phase with a mean half-life of about ~ 98 hrs. The half-life was lower in our patient population than reported in literature (~4 vs 20 days). In our study, we collected blood samples out to 7 days post administration and the data analysis was conducted using non-compartmental methods. However, the previously reported pharmacokinetic study sampled out much longer than our study (out to 12 months) [31]. It is clear after approximately 200 hours, there is a prolonged half-life for moxidectin, however, we did not sample out that long. The elimination half-life calculation is impacted by the duration of plasma sampling. The obtained MOX mean $AUC_{0-\infty}$ value shows a similar drug exposure in LF patients to what is previously reported in healthy individuals (3405 ng*hr/mL vs 3387 ng*hr/mL, respectively); however, the mean peak plasma concentration $C_{max}$ was ~ two fold higher in our patient population than reported in healthy individuals (127 ng/mL vs 58.90 ng/mL). The total apparent clearance (Cl/F) of MOX was similar to reported values in healthy individuals, but the apparent volume of distribution ($V_D/F$) was lower in our population (410.8 L vs 2421 L) [19,20,31, 32]. The possible explanation is that in our study, MOX was administered with food compared to fasting status that reported values in healthy

**Table 2. Pharmacokinetic parameters of the study drugs when administered in IDA, IA, MOXDA and MOXA study arms.**

| Drugs | ARMS | $C_{max}$ (ng/mL) | $T_{max}$ (hr) | Half-life $(t_{1/2})$(h) | $AUC_{0-t}$ (ng*hr/mL) | $AUC_{0-\infty}$ (ng*hr/mL) | $V_{z/F}$ (L) | Cl/F (L/hr) | Dose Normalized $C_{max}$ (ng/mL) | Dose Normalized $AUC_{0-t}$ (ng*hr/mL) | Dose Normalized $AUC_{0-\infty}$ (ng*hr/mL) |
|---|---|---|---|---|---|---|---|---|---|---|---|
| ALB | IDA | 40.2 (2.4–167.8) | 4.1 (3–6) | 6.9 (1.5–40.9) | 202.9 (13.7–643.7) | 204.1 (14–644.8) | 85681.8 (4512.2–773481) | 8211.1 (620.4–28592.2) | 24.3 (1.7–95.3) | 123.4 (9.6–365.6) | 124.2 (9.9–366.2) |
| | IA | 38.4 (4.3–179.1) | 3.5 (2–6) | 8.7 (3–19.3) | 189.2 (14.4–1270.1) | 190.8 (14.8–1270.9) | 63366.1 (4019.3–280885.5) | 6205.1 (314.8–27059.4) | 24.2 (2.9–96.7) | 115.2 (9.8–685.9) | 116.2 (10–686.3) |
| | MOXDA | 25.4 (2.5–121.3) | 3.6 (2–6) | 10.1 (1.8–24.3) | 134 (21.9–537.4) | 136 (22.4–537.7) | 90516.6 (2318.3–501395.3) | 7038.7 (743.9–17827.7) | 15.7 (1.4–87.2) | 81.3 (12.9–386.4) | 82.4 (12.9–386.6) |
| | MOXA | 66.1 (2.2–320.2) | 3.3 (2–6) | 10 (2.2–27.3) | 290 (11.2–1231.4) | 293.3 (12.6–1252.3) | 144442.6 (3791.1–621813.6) | 8586.8 (319.4–31799.4) | 37.9 (1.3–165.9) | 170.2 (7.1–637.9) | 172.1 (8–648.7) |
| ALB-OX | IDA | 469.4 (132.4–1475.8) | 5.6 (4–6) | 7.9 (5–12.5) | 5693.4 (1832.6–17514.9) | 5721.6 (1840–17536.8) | 1229.8 (234.9–2788.5) | 107.8 (22.8–217.4) | 284.8 (93.3–814.7) | 3492.8 (1254.9–9668.2) | 3512.5 (1264.8–9680.3) |
| | IA | 422.7 (197.3–1471.6) | 4.3 (3–6) | 8.7 (5.9–12.9) | 5227.7 (2221.3–20615.4) | 5251.3 (2225.5–20642.2) | 1328.1 (198.9–2722.9) | 106.7 (19.4–179.7) | 268.3 (123.9–794.7) | 3286.5 (1350.2–11132.3) | 3302.3 (1355.5–11146.8) |
| | MOXDA | 334.3 (125.7–571.3) | 5.3 (4–6) | 8.6 (4–14) | 4326.6 (1777.1–7796.6) | 4372.2 (1777.1–7981.2) | 1344.7 (522.5–3980.7) | 112 (50.1–221.9) | 200 (63.5–410.7) | 2568 (897.4–5605.8) | 2595.8 (897.4–5738.5) |
| | MOXA | 495.3 (115.1–1165) | 4.6 (3–6) | 9.5 (6.8–12.7) | 6845.2 (1521.2–16517.8) | 6880.8 (1523–16543.5) | 1436.6 (241.1–3775.1) | 100 (24.2–262.7) | 294 (67.3–615.1) | 4057.5 (1102.9–8556.2) | 4078.9 (1104.2–8569.5) |
| ALB-ON | IDA | 29.1 (7.9–86.5) | 5.5 (4–6) | 9.3 (5.6–20.5) | 350.9 (115.7–1103.3) | 356.6 (116.1–1111.5) | 22232.5 (3269.7–52154.8) | 1641.6 (359.9–3445.1) | 17.6 (5.5–47.8) | 213.3 (79.2–609) | 217.1 (79.5–613.5) |
| | IA | 22.5 (8.7–59.6) | 4.9 (3–6) | 9.4 (6–13.3) | 289.3 (70.6–1091) | 297.2 (71.8–1099.9) | 25460.2 (3238.4–67992.7) | 1868.3 (363.7–5570.5) | 14.4 (5.2–32.2) | 182.2 (42.4–589.1) | 187.3 (43.2–593.9) |
| | MOXDA | 21.6 (9.2–35.3) | 6.1 (4–12) | 9.7 (5.1–17.8) | 280.8 (136.1–647.4) | 286.6 (139.2–648.4) | 24027.5 (6707.5–60333) | 1658.2 (616.9–2874.4) | 13 (6.1–20.7) | 170.5 (87.4–465.5) | 174 (89.4–466.2) |
| | MOXA | 30.6 (8–80.6) | 5.6 (3–12) | 10.5 (6.3–13.7) | 501.7 (143.7–1488.9) | 507.4 (147.7–1496.2) | 19089.6 (3128.4–50357.1) | 1182.8 (267.3–2708.9) | 18.5 (4.8–42.6) | 297.2 (86.8–771.3) | 300.8 (89.2–775) |
| DEC | IDA | 1167.3 (750.6–1408.5) | 3.9 (3–6) | 11.1 (9–21.2) | 20521.1 (10560.6–27856.5) | 20899.2 (10618.4–28750.6) | 143.2 (91–216.7) | 9.1 (6.9–14.4) | 63.4 (50.9–83.8) | 1214.6 (794.4–1523.9) | 21308.2 (11177.4–28869.3) |
| | MOXDA | 1175.8 (837.4–1581.7) | 4.4 (2–12) | 10.5 (8.3–12.6) | 19452.2 (11971.7–25281.9) | 19610.3 (12008.9–25505.9) | 140.5 (93.1–173.5) | 9.5 (6–13.6) | 59.4 (50–72.5) | 1177 (894.6–1623.3) | 19544.9 (12412.9–28740.7) |
| IVM | IDA | 52.3 (25.9–103.1) | 6.5 (3–12) | 48.7 (21.9–85.9) | 1670.9 (983.8–3389.4) | 1916.7 (1039.2–4537.4) | 489.1 (205.3–986) | 7.3 (3.3–11.6) | 63.4 (50.9–83.8) | 52.6 (27.5–95.9) | 1690.9 (885.5–3222.2) |
| | IA | 61 (23.7–160.6) | 6.1 (4–12) | 61 (13–177.6) | 1532.2 (538.3–3150.4) | 1895 (692.3–5900.1) | 648.8 (321.5–1174.1) | 9.5 (2.5–21.7) | 65.1 (54–79) | 59.7 (26.1–153.8) | 1505.1 (606.5–3318.4) |

(*Continued*)

**Table 2.** (Continued)

| Drugs | ARMS | $C_{max}$ (ng/mL) | $T_{max}$ (hr) | Half-life $(t_{1/2})$(h) | $AUC_{0-t}$ (ng*hr/mL) | $AUC_{0-\infty}$ (ng*hr/mL) | $V_{z/F}$ (L) | Cl/F (L/hr) | Dose Normalized $C_{max}$ (ng/mL) | Dose Normalized $AUC_{0-t}$ (ng*hr/mL) | Dose Normalized $AUC_{0-\infty}$ (ng*hr/mL) |
|---|---|---|---|---|---|---|---|---|---|---|---|
| MOX | MOXDA | 136.7 (33.9–476.5) | 5 (3–6) | 72.7 (24.4–178) | 2712 (1080–7764.5) | 3442.4 (1441.2–11375.5) | 307.9 (103–860.8) | 3.3 (0.7–5.6) | 103.7 (24.8–314.5) | 2088.5 (789.7–5124.6) | 2653.2 (993.5–7507.8) |
|  | MOXA | 117.7 (30.5–242.4) | 5.4 (4–6) | 98.5 (36.6–162.8) | 2614.1 (627–5124.5) | 3367 (742.2–7479.5) | 513.7 (78.6–1088.5) | 3.8 (1.1–10.8) | 87.2 (20.8–188.6) | 1960.5 (464–3801.7) | 2530.8 (549.2–5933.8) |

Data presented are the Mean (range) values for each pharmacokinetic parameter.

$T_{1/2}$ terminal half-life, $T_{max}$ time of maximum plasma concentration, $C_{max}$ maximum plasma concentration, AUC area under the concentration-time curve, Vz/F apparent volume of distribution, CL/F apparent clearance.

ALB-OX, albendazole sulfoxide; ALB-ON, albendazole sulfone; DEC, diethylcarbamazine; IVM, ivermectin; MOX, Moxidectin.

IDA, three-drug combination (DEC 6mg/kg+ IVM 200ug/kg + ALB 400mg); IA, two drug combination (IVM 200µg/kg + ALB 400mg); MOXDA, three drug combination (DEC 6mg/kg+ MOX 8mg + ALB 400mg); and MOXA, two-drug combination (MOX8 + ALB 400mg).

individuals. Bioavailability and $C_{max}$ of MOX is increased when taking the drug with food, especially with a high-fat diet [20].

Results from this study show that the addition of MOX does not affect DEC or ALB drug levels or their PK parameters compared to IVM combination in LF patients. Our findings suggest that the co-administration of MOXDA and MOXA vs. IDA or IA regimen for LF did not result in significant alterations in the drug concentrations of co-administered agents. The plasma drug concentration profile in Bancroftian filariasis-infected subjects is similar to that in healthy volunteers and subjects with *O. volvulus* infections [19,20,31,32]. The exposure ($C_{max}$) for MOX in MOXA arm (mean, 117.7 ng/mL) v/s MOXDA (mean, 136.7 ng/mL) and IVM for IA (mean, 61 ng/mL) v/s IDA (mean, 52.3 ng/mL) is also not significantly altered

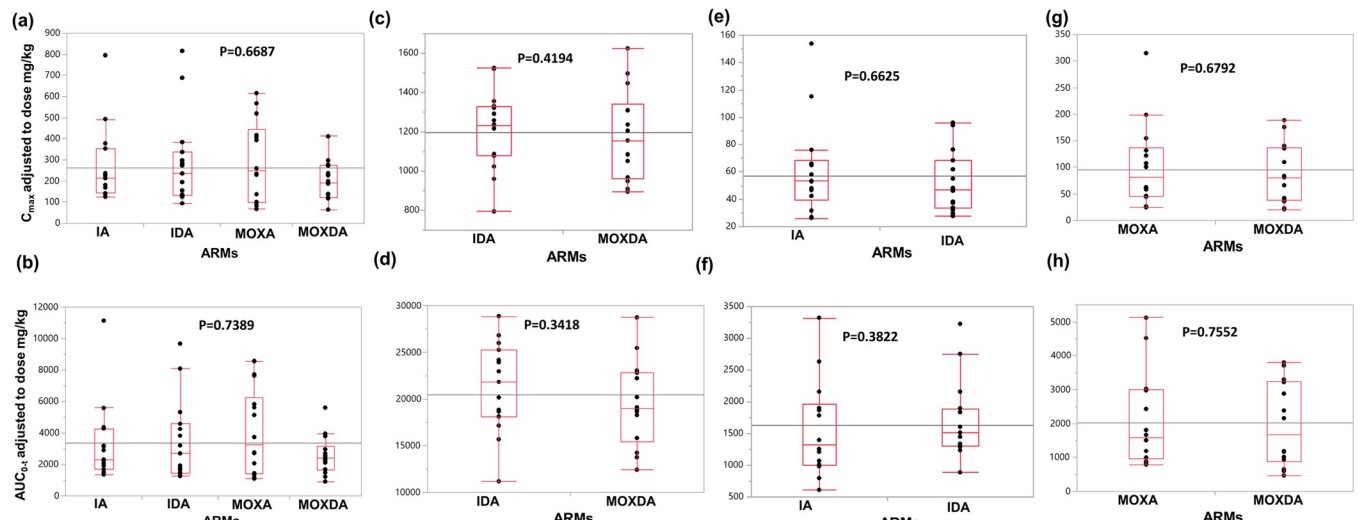

**Fig 2.** Distribution of dose normalized Cmax and $AUC_{0-t}$ of ALB-OX (a & b) DEC (c & d), IVM (e & f) and MOX (g & h) by study arm. The box plots indicate the 25% to 75% percentiles, while the error bars represent the 5% and 95% percentiles. Individual data points are indicated by solid dots. The overall median value for both groups is indicated by the horizontal lines within each box. Solid line between both groups indicates grand mean value. Significance was assessed using the Kruskal-Wallis test and all P values were > 0.05. (IDA, n = 15, IA, n = 15, MOXDA, n = 14, MOXA, n = 14).

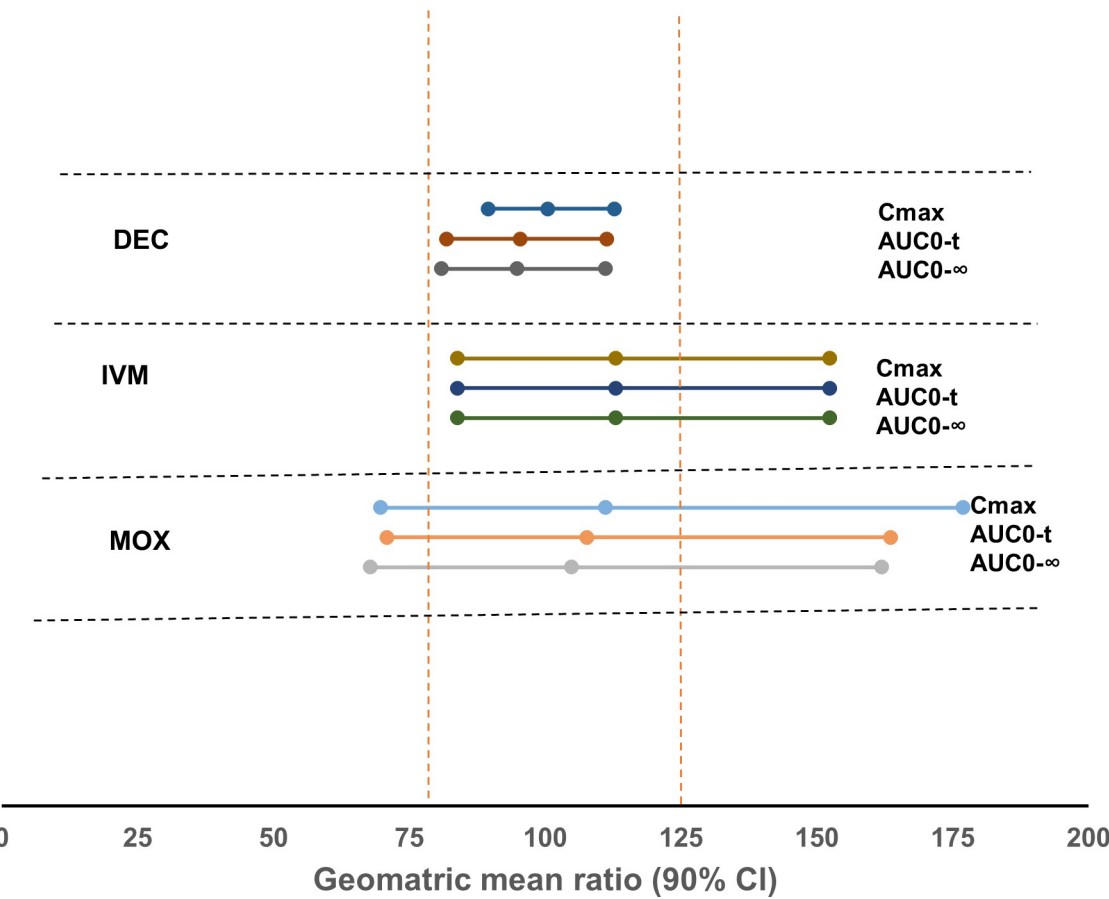

**Fig 3. Forest plots of the geometric mean ratios (±90% confidence intervals [CI]) of the drug administered for the experimental regimen and the reference regimens for logarithmically transformed $C_{max}$ and $AUC_{0-t}$ and $AUC_{0-\infty}$.**

showing not a huge impact of DEC combination in both MOXA and IA, 2-drug treatment regimen.

The GMR values of PK parameters for IVM, DEC, and ALB-OX metabolites, were not significantly altered by the co-administration with MOX in different treatment arms, and values were in line with previous studies [8,10,11]. These results suggest that MOX has no clinically relevant effect on the PK of DEC or ALB. Moreover, there was no apparent change in the PK for MOX when administered in this combination two or three-drug regimens. There was considerable variability in plasma ALB and IVM drug levels among individuals, as has been previously reported [10,12]. Previous studies evidenced the favorable PK safety profile of combination treatments for neglected tropical diseases (NTDs) viz. azithromycin and IVM, IVM and ALB, IDA drugs, and IVM, ALB and azithromycin [8,10,11,33]. MOX has a large apparent volume of distribution and low oral clearance, resulting in long terminal $T_{1/2}$ (mean values of ~ 98 hrs in this study) in *W. bancrofti*-infected individuals.

A variety of literature studies revealed that MOX is a poor substrate for P-glycoprotein (P-gp) transporters and is mostly excreted via a P-gp-independent pathways into the intestine [34,35]. *In vitro* studies using mammalian liver microsomes and S9 suggested that MOX is a limited substrate of CYP-metabolism (cytochrome P450 3A and cytochrome P450 2B [26,36], producing a small number of hydroxylated metabolites. There is no evidence for non-CYP-mediated metabolism. In humans, MOX is minimally metabolized within the body and does

not affect the pharmacokinetics of midazolam, a sensitive CYP3A4 substrate [37]. MOX is excreted mostly unmetabolized in feces [38]. These results lead to the conclusion that MOX is not likely to cause clinically relevant cytochrome P450-related drug–drug interactions.

We found no effect on MOX PK in different study arms and drug exposure parameters ($AUC_{0-t}$ and $C_{max}$). $T_{max}$, $C_{max}$, $AUC_{0-t}$, and CL/F obtained in the current study are also consistent with those reported in previous studies [19,20,31,32]. These PK results suggest moxidectin co-administration does not alter the pharmacokinetics of DEX or albendazole in LF treatment programs.

Additional studies evaluating covariates that may account for high variability are warranted to better understand and validate MOX PK in LF patients. The benefits of MDA integration include increased coverage and geographic reach of national NTD programs, whilst achieving financial and programmatic savings. Integrated MDAs will be of particular value in countries where these diseases are co-endemic and where the cost of individual MDA is particularly high compared to other settings (34).

## Supporting information

**S1 Table. Pharmacokinetic parameters of the study drugs when administered in IDA, IA, MoxDA and MoxA study Arms.**
(DOCX)

## Acknowledgments

Participants who willingly participated in this study are acknowledged and thanked for their contribution. We thank MDGH for donating the moxidectin used in this study.

## Author Contributions

**Conceptualization:** Catherine Bjerum, Peter U. Fischer, Gary J. Weil, Christopher L. King, Philip J. Budge.

**Data curation:** Allassane F. Ouattara, Benjamin G. Koudou, Toki P. Gabo, Abdoulaye Meïté.

**Formal analysis:** Veenu Bala, Daryl J. Murry.

**Funding acquisition:** Philip J. Budge.

**Methodology:** Yashpal S. Chhonker, Allassane F. Ouattara, Benjamin G. Koudou.

**Software:** Yashpal S. Chhonker, Veenu Bala, Daryl J. Murry.

**Supervision:** Daryl J. Murry.

**Validation:** Yashpal S. Chhonker.

**Writing – original draft:** Yashpal S. Chhonker, Veenu Bala, Abdullah Alshehri, Daryl J. Murry.

**Writing – review & editing:** Yashpal S. Chhonker, Catherine Bjerum, Veenu Bala, Allassane F. Ouattara, Benjamin G. Koudou, Toki P. Gabo, Abdullah Alshehri, Abdoulaye Meïté, Peter U. Fischer, Gary J. Weil, Christopher L. King, Philip J. Budge, Daryl J. Murry.

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
