## [Decision Letter · Decision Letter 0]

27 May 2023

Dear Pharm.D Murry,

Thank you very much for submitting your manuscript "Pharmacokinetics of Moxidectin Combined with Albendazole or Albendazole plus Diethylcarbamazine for Bancroftian Filariasis" for consideration at PLOS Neglected Tropical Diseases. As with all papers reviewed by the journal, your manuscript was reviewed by members of the editorial board and by several independent reviewers. The reviewers appreciated the attention to an important topic. Based on the reviews, we are likely to accept this manuscript for publication, providing that you modify the manuscript according to the review recommendations. 

The Academic Editor apologizes for the delay in completing this review. In light of the challenge of finding reviewers who have experience in pharmacokinetics and LF, I elected to provide a second review, as I have experience in this area.

The responsive reviewer found considerable merit in the manuscript to the extent that the data presented are generally consistent with a conclusion that marked drug-drug interactions do not occur among the three medicines used in triple therapy for LF. However, the reviewer noted that the trial design is not entirely appropriate for a direct DDI study; although I agree, I do not feel that this distracts from the conclusions. Nonetheless, it should be ackowledged in the text. Like the responsive reviewer, I am somewhat concerned by the discrepancy in plasma half-lives observed here compared to previous reports, and this should also be addressed by the authors in the Discussion.

Lastly, the authors should deposit the raw data in an accessible database, fully anonymized, and provide that location in the text.

Like the responsive reviewer, I believe this manuscript will be a valuable addition to the literature and look forward to receipt of an appropriately revised version.

Sincerely,

Timothy G. Geary, PhD

Guest Editor

Cinzia Cantacessi

Section Editor

The Academic Editor apologizes for the delay in completing this review. In light of the challenge of finding reviewers who have experience in pharmacokinetics and LF, I elected to provide a second review, as I have experience in this area.

The responsive reviewer found considerable merit in the manuscript to the extent that the data presented are generally consistent with a conclusion that marked drug-drug interactions do not occur among the three medicines used in triple therapy for LF. However, the reviewer noted that the trial design is not entirely appropriate for a direct DDI study; although I agree, I do not feel that this distracts from the conclusions. Nonetheless, it should be ackowledged in the text. Like the responsive reviewer, I am somewhat concerned by the discrepancy in plasma half-lives observed here compared to previous reports, and this should also be addressed by the authors in the Discussion.

Lastly, the authors should deposit the raw data in an accessible database, fully anonymized, and provide that location in the text.

Like the responsive reviewer, I believe this manuscript will be a valuable addition to the literature and look forward to receipt of an appropriately revised version.

Reviewer's Responses to Questions

**Key Review Criteria Required for Acceptance?**

**Methods**

-Are the objectives of the study clearly articulated with a clear testable hypothesis stated?

-Is the study design appropriate to address the stated objectives?

-Is the population clearly described and appropriate for the hypothesis being tested?

-Is the sample size sufficient to ensure adequate power to address the hypothesis being tested?

-Were correct statistical analysis used to support conclusions?

-Are there concerns about ethical or regulatory requirements being met?

Reviewer #1: The objective of the study “…is to determine whether the incorporation of moxidectin (MOX) into MDA regimens for LF will alter the pharmacokinetics of co-administered drugs”. However, this is a sub study and this was not the primary objective of the clinical trial. 

In terms of moxidectin’s risk of drug:drug interactions (DDIs), these were formally evaluated as part of its development by Wyeth/Pfizer, MDGH and the US FDA. Based on the metabolic pathways, transporters, and previous data with midazolam as a probe (Korth-Bradley et al, 2014), the risk of DDIs with this combination of compounds was negligible based on in silico modelling. While the paper describes data that are supportive of this position, this was not prospectively designed as a drug:drug interaction study (i.e clinical DDI studies compare substrate concentrations in the absence and presence of a perpetrator or victim drug). As such, negative findings (in terms of not identifying a DDI) from retrospective studies such as this (i.e not prospectively defined solely for the purpose of DDI evaluation) have limited value and cannot inform regulatory authorities. Recommend that the authors review FDA Guidance for Industry https://www.fda.gov/media/134581/download to help understand the deficiencies between this design and a DDI study. 

A comment on how patients were selected for the pharmacokinetic sub study and how similar or different they were to the broader population in the clinical trial would also be helpful.

**Results**

-Does the analysis presented match the analysis plan?

-Are the results clearly and completely presented?

-Are the figures (Tables, Images) of sufficient quality for clarity?

Reviewer #1: The pharmacokinetic results of moxidectin and ivermectin are different from previously published data - ivermectin T1/2 is 15 hours in the US FDA PI and moxidectin’s is between 20 and 40 days. Ivermectin’s T1/2 was significantly longer than published data and moxidectin’s significantly shorter. Data from doi:10.1371/journal.pntd.0010005 should be the primary comparison used in the discussion. Performance of the assays used should be commented upon - how the assays were validated should be commented on. Also, their performance (e.g. range of quantation, inter day, inter operator performance parameters should be given). The number of available samples at each time point should be noted at some place in the paper - were these analyses based on available samples or on patients with complete sets? Also, were the samples all compliant with sampling windows and if not, how many were deviant from prospectively defined windows? How close was dosing relative to food intake?

**Conclusions**

-Are the conclusions supported by the data presented?

-Are the limitations of analysis clearly described?

-Do the authors discuss how these data can be helpful to advance our understanding of the topic under study?

-Is public health relevance addressed?

Reviewer #1: The conclusions are supported but there is a tendancy to overstate the value of the data. This study was not the designed by standard DDI methodologies to evaluate the question. The outlying data from this study compared to all other pharmacokinetic data on moxidectin and ivermectin warrant comment - it leads to the differences being due to disease questions or study execution/assay questions. The underlying message remains that the data do not generate concerns about the use of these agents together.

**Editorial and Data Presentation Modifications?**

Reviewer #1: It should be noted that DEC can result in life threatening system reactions and not just ocular reactions in onchocerciasis patients. 

It should say that this is a sub study.

**Summary and General Comments**

Reviewer #1: The data are of value. The comments on the experiment are already noted but the authors are to be congratulated on executing this trial.

PLOS authors have the option to publish the peer review history of their article (what does this mean?). If published, this will include your full peer review and any attached files.

Reviewer #1: No

Figure Files:

Data Requirements:

Reproducibility:

References

---

## [Editor Report · Decision Letter 1]

1 Aug 2023

Dear Pharm.D Murry,

We are pleased to inform you that your manuscript 'Pharmacokinetics of Moxidectin Combined with Albendazole or Albendazole plus Diethylcarbamazine for Bancroftian Filariasis' has been provisionally accepted for publication in PLOS Neglected Tropical Diseases.

Best regards,

Timothy G. Geary, PhD

Guest Editor

Cinzia Cantacessi

Section Editor

The authors have resolved the concerns expressed by the reviewer as part of the process of evaluation, and have as a result improved the clarity and quality of the manuscript. As a scientist with some expertise in this area, the Associate Editor believes that the manuscript meets the standards associated with publication in PLoS-NTDs.

---

## [Editor Report · Acceptance letter]

20 Aug 2023

Dear Pharm.D Murry,

We are delighted to inform you that your manuscript, "Pharmacokinetics of Moxidectin Combined with Albendazole or Albendazole plus Diethylcarbamazine for Bancroftian Filariasis," has been formally accepted for publication in PLOS Neglected Tropical Diseases.

Best regards,

Shaden Kamhawi

co-Editor-in-Chief

Paul Brindley

co-Editor-in-Chief
